# Assessment of Indoor Air Quality for Group-Housed Macaques (*Macaca* spp.)

**DOI:** 10.3390/ani12141750

**Published:** 2022-07-07

**Authors:** Annemiek Maaskant, Isabel Janssen, Inge M. Wouters, Frank J. C. M. van Eerdenburg, Edmond J. Remarque, Jan A. M. Langermans, Jaco Bakker

**Affiliations:** 1Animal Science Department, Biomedical Primate Research Centre, 2288 GJ Rijswijk, The Netherlands; langermans@bprc.nl (J.A.M.L.); bakker@bprc.nl (J.B.); 2Department Population Health Sciences, Animals in Science & Society, Faculty of Veterinary Medicine, Utrecht University, 3584 CM Utrecht, The Netherlands; 3Department Population Health Sciences, Farm Animal Health, Faculty of Veterinary Medicine, Utrecht University, 3584 CM Utrecht, The Netherlands; da.isabeljanssen@gmail.com (I.J.); f.j.c.m.vaneerdenburg@uu.nl (F.J.C.M.v.E.); 4Department Population Health Sciences, Institute for Risk Assessment Sciences, Faculty of Veterinary Medicine, Utrecht University, 3584 CM Utrecht, The Netherlands; i.wouters@uu.nl; 5Virology Department, Biomedical Primate Research Centre, 2288 GJ Rijswijk, The Netherlands; remarque@bprc.nl

**Keywords:** non-human primate, dust, ammonia, endotoxin, humidity, ventilation rate, air quality, airborne fungi

## Abstract

**Simple Summary:**

Indoor air quality is strongly associated with animal health and wellbeing. Therefore, animal enclosures must be consistently and sufficiently ventilated to provide for the health and well-being of animals and caretakers. Although there are several publications concerning assessments and effects of suboptimal air quality on rodents, no publications are available on group-housed non-human primates and the exposure of caretakers to inhalable dust and endotoxins. The indoor air quality of group-housed macaques at the Biomedical Primate Research Center (Rijswijk, the Netherlands) was assessed to identify possible problems regarding air circulation and the concentrations of inhalable dust, endotoxin, ammonia, fungi, temperature and relative humidity in the indoor environment. In addition, the exposure to inhalable dust and endotoxins of caretakers was evaluated. The observed values for these air quality parameters, measured at fixed locations in the animal enclosures, did not exceed the proposed human threshold limit values (TLV). However, caretakers were exposed to higher levels than the animals likely due to nature of their tasks. This study provides practical tools that can be used to improve the indoor air quality in group-housed macaques. Moreover, the results show that the exposure of caretakers to inhalable dust and endotoxins during daily work routines should be reduced.

**Abstract:**

Indoor Air Quality (IAQ) is strongly associated with animal health and wellbeing. To identify possible problems of the indoor environment of macaques (*Macaca* spp.), we assessed the IAQ. The temperature (°C), relative humidity (%) and concentrations of inhalable dust (mg/m^3^), endotoxins (EU/m^3^), ammonia (ppm) and fungal aerosols were measured at stationary fixed locations in indoor enclosures of group-housed rhesus (*Macaca mulatta*) and cynomolgus macaques (*Macaca fascicularis*). In addition, the personal exposure of caretakers to inhalable dust and endotoxins was measured and evaluated. Furthermore, the air circulation was assessed with non-toxic smoke, and the number of times the macaques sneezed was recorded. The indoor temperature and relative humidity for both species were within comfortable ranges. The geometric mean (GM) ammonia, dust and endotoxin concentrations were 1.84 and 0.58 ppm, 0.07 and 0.07 mg/m^3^, and 24.8 and 6.44 EU/m^3^ in the rhesus and cynomolgus macaque units, respectively. The GM dust concentrations were significantly higher during the daytime than during the nighttime. Airborne fungi ranged between 425 and 1877 CFU/m^3^. Personal measurements on the caretakers showed GM dust and endotoxin concentrations of 4.2 mg/m^3^ and 439.0 EU/m^3^, respectively. The number of sneezes and the IAQ parameters were not correlated. The smoke test revealed a suboptimal air flow pattern. Although the dust, endotoxins and ammonia were revealed to be within accepted human threshold limit values (TLV), caretakers were exposed to dust and endotoxin levels exceeding existing occupational reference values.

## 1. Introduction

In laboratory animal science, a good management program provides optimal housing and care to ensure high-standard animal models, laboratory animal welfare and the quality of animal research [1]. However, laboratory animal care and use guidelines are lacking scientific evidence regarding ventilation rates and recommendations to ensure optimal IAQ for non-human primates (NHP). Although there are several publications concerning assessments and effects of suboptimal air quality in rodents, no publications are available on group-housed NHP and the exposure of caretakers to inhalable dust and endotoxins [2,3,4,5].

Many care programs—such as the European directive 2010/63/EU and the Guide for the Care and Use of Laboratory Animals [6]—adopt the general recommendation of 10 to 15 air changes per hour for animal rooms, regardless of housing type and conditions [1]. Yet, others proposed that, depending on the housing type and conditions, a minimum of six air changes per hour could also be sufficient [7,8].

Appropriate ventilation supplies adequate fresh air, maintains optimal temperature and humidity and dilutes gaseous (e.g., ammonia) and particulate air pollutants, such as (inhalable) dust and its contaminants. Inhalable organic dust particles (<100 µm) originate from plant fragments, skin scales, fur and microbes, such as bacteria and fungi [9]. The outer cell wall of Gram-negative bacteria contains lipopolysaccharide structures, also known as endotoxins [10,11]. Endotoxins are ubiquitous in the environment; however, in occupational environments and agricultural settings—such as animal stables—the concentrations are higher [12,13]. 

In addition, ammonia is generated by bacterial activity on unabsorbed nutrients and urea in animal feces and urine. Ammonia emission is correlated with environmental temperature and humidity [14]. In both humans and animals, high concentrations of endotoxins and ammonia are reported to be associated with acute and chronic respiratory symptoms [12,13,15]. Therefore, next to sufficient hygiene measures, animal enclosures must be consistently and sufficiently ventilated to ensure the health and well-being of animals and caretakers.

Over the last decades, it has been recognized that animal health and welfare must be improved for laboratory animals. Guidelines are continuously updated, and laboratory animal housing standards for NHP improved over time accordingly. Currently, at the Biomedical Primate Research Centre (BPRC, Rijswijk, the Netherlands), the macaques in the breeding facility are housed in spacious and stimulating cages comprising both indoor and outdoor enclosures. However, no data are available for this housing type regarding IAQ parameters, e.g., inhalable dust, endotoxins, ammonia and fungal aerosols. 

To identify possible problems of the indoor environment, we assessed the IAQ in two group-housed macaque units (*Macaca mulatta* and *Macaca fascicularis*). Second, we evaluated the exposure to inhalable dust and endotoxins of caretakers during their daily work. The results were compared to existing human TLV. Third, we assessed whether animal activity influenced the IAQ parameters and whether the observed number of sneezes during the assessment could be a non-invasive predictor for air quality. The newly acquired knowledge can be beneficial to improve the IAQ in macaque colonies and the health risks to the caretakers.

## 2. Materials and Methods

### 2.1. Animals, Husbandry and Housing

The study groups in this research consisted of both rhesus macaques (*Macaca mulatta*) and cynomolgus macaques (*Macaca fascicularis*) from the breeding colonies of the BPRC. All procedures, husbandry and housing performed in this study were in accordance with the Dutch laws on animal experimentation and the regulations for animal handling as described in European directive 2010/63/EU. BPRC is accredited by AAALAC International. Before the start of this observational study, approval was obtained by the institutional animal welfare body (IvD 022A).

In this study, four groups were selected based on housed species, occupancy rate and comparability regarding the location inside the units. To evaluate the IAQ for the two macaque species housed at the BPRC, a rhesus macaques unit (RMU) and cynomolgus macaques unit (CMU) were selected. These units consisted of two separate animal rooms, and each room consisted of a passageway for the caretakers and the animal enclosures. These indoor enclosures were divided into compartments by concrete walls with passages for the animals.

The indoor enclosures were 2.85 m high and consisted of two (CMU) or three (RMU) connected compartments, with a floor surface of 25 m^2^ each. A single enclosure housed a multi-generational group consisting of males and females. The animal details are summarized in Table 1. In the RMU, two of the three compartments were directly connected to the outdoor enclosures, whereas both compartments in the CMU were connected to the outdoor enclosures. The indoor and outdoor enclosures were freely accessible for the animals by passing hatches with flexible strip curtains that separated the areas.

In both units, the front of the enclosures consisted of galvanized steel fencing with 5 × 5 cm spot-welded mesh wire. Although the size and height of the compartments were identical in both units, there were some differences in the design of the front fencing. First, the balcony was located on different heights for the RMU and CMU. Second, the design and location of the sliding doors differed between the units, including the location and size of support beams. Third, in CMU, an additional parallel fence was present 2.5 m from the front. Last, in CMU, a U-profile (4 × 15 cm) was secured on the front fence to prevent the cynomolgus macaques from touching the control wires of the indoor animal passages (Figure 1).

The floors in the indoor enclosures were provided with wood fiber bedding (Lignocel^®^ 3–4, JRS, Rosenberg, Germany). Standard environmental enrichment in these enclosures consisted of several climbing structures, beams, fire hoses, car tires and sitting platforms to stimulate natural behavior and free access to the outdoor enclosures. Drinking water was ad libitum available via automatic water dispensers. The animals were fed commercial monkey pellets (Ssniff, Soest, Germany) and daily limited amounts of vegetables, fruits or grain mixtures were offered.

Cleaning and enrichment was performed according to standardized protocols. The bedding of the indoor enclosures were cleaned out weekly. High-pressure water cleaning, including disinfection (Anistel Surface disinfectant, Tristel Solutions Limited, Cambridgeshire, UK), was performed monthly. Following disinfection, the enclosures were rinsed with clean water, and the floor was wiped dry. After allowing for a 30–40 min air dry period, approximately 31 kg of wood fiber was provided as bedding in each compartment after each cleaning procedure. Subsequently, enrichment items were provided, e.g., cardboard boxes filled with shredded paper and some mixed grains [16]. During the study, all groups received the same enrichment items.

Outdoor air entered through the air handling unit BD-7 (VBW clima automatic, Gdynia, Poland) on RMU and LBK 02 AH AT4 (AL-KO luchttechniek b.v. Roden, Netherlands) on CMU and flowed through Hi -FLO -HFGS F7 ISO 16890 ePM1 70% filters (Camfil, Ede, Netherlands) into the duct system of the air ventilation system. The filters prevent 70% of particles of <1 μm in size from passing. The duct system was attached directly under the ceiling of the corridors in the animal rooms. In front of each compartment, one ventilation inlet was located, provided with a vent grille (22 × 60 cm) to guide the airflow into the enclosures.

In the RMU, the exhausts (40 × 80 cm) were located in the middle of the wall opposite to the enclosures and right under the ceiling and above the floor (Figure 2 and Figure 3). The ventilation exhausts (31 × 31 cm) in the CMU were located on the left and right side of each room right under the ceiling, opposite to the cages as well. A ventilation rate of six air changes per hour was considered sufficient due to the large cubic capacity of the animal rooms and the relatively low occupancy rates. Furthermore, the outdoor enclosures were freely accessible to the animals during both day and night.

The minimum indoor temperature was controlled by heating the air inside the air handling unit and a radiant heating system inside the walls of the compartments. The minimum indoor temperature was set to 18 °C in the RMU and 21 °C in the CMU, respectively. Due to the maritime climate in the Netherlands combined with the accessibility to the outdoor enclosures, the units were designed without a cooling and humidity control system.

### 2.2. Study Design

The study was performed from July to September 2020. The equipment was placed, i.e., the air was sampled, in two indoor compartments of each study group (Figure 2). Dust, endotoxin and ammonia samples were simultaneously obtained for five days for approximately six hours a day (435 ± 12 min). The study days were selected based on the cleaning schedules of the animal rooms (Appendix A). A similar interval of days after cleaning was aimed for; however, this interval was not synchronized between the units.

To correct for the variability in the natural occupancy rate in the compartments during the day, two night measurements were included. Camera surveillance confirmed that the animals stayed indoors during the nights. The night samples were obtained for approximately 10 h a night (625 ± 155 min).

The temperature and relative humidity were measured alongside the previously mentioned parameters with a recording interval of 10 min. All measurements were performed simultaneously in RMU and CMU. The fungal sampling was performed on one separate day.

To protect the measuring equipment against the inquisitive macaques, a cage construction with a mesh wire of 5 × 5 cm, with an additional mesh wire of 1 × 1 cm around the equipment, was designed ensuring a free airflow (Figure 4). The construction measured 42.5 × 37.5 × 125.0 cm and was secured to the ceiling. The air quality was sampled approximately 1.6 m above the cage floor and 1.25 m from the ceiling in the middle of two compartments of the indoor enclosures (Figure 1).

During the study days, animal activities were ad libitum live observed and recorded by two observers in four sessions of 30 min for each group. Two sessions were performed in the morning and two in the afternoon. The observers were randomly assigned to a unit and to a group to start the first observation, and subsequent group observations were alternately performed. The defined and noted activities were: (1) foraging, (2a) play terrestrial, (2b) play arboreal, (3) rest and (4) aggressive interaction (Table 2). The activities were selected based on their potential to influence IAQ parameters, e.g., manipulation of the wood fiber bedding could potentially increase the measured dust concentration.

The number of animals present in the two compartments and the activities were recorded with a sample interval of five minutes. During the observational sessions, mothers with their offspring in the ventro–ventral position were counted as one animal. In addition, the numbers of sneezes were scored as a non-invasive health indicator for IAQ simultaneously with the animal activities.

### 2.3. Sampling Techniques

#### 2.3.1. Dust and Endotoxin

Inhalable dust was collected on 37 mm Whatman^®^ GF/A glass microfiber filters (Whatman International Ltd., Maidstone, UK) using Gillian GilAir-5 pumps (Gillian, Sensidyne, Clearwater, FL, USA) connected with a flexible tube a conical inhalable sampler (CIS) sampling head (JS Holdings, Stevenage, UK) in which the filter was mounted. Inhalable dust particles perched on the filter after activating the GilAir-5 pump. During all experimental days, a control filter was present. At the start of the sampling day, the pumps were calibrated at a flow rate of 3.5 L/min using a rotameter (Brooks Instruments, Hatfield, Pennsylvania) and repeated at the end of a sampling day. Immediately after collection, dust filters were stored at −20 °C until further processing. The dust concentrations (mg/m^3^) were assessed as described previously [17]. All filters were pre- and post-weighed at the same time in an acclimated room on an analytical balance with 0.01 mg readability. The acclimated room maintained a constant temperature, humidity and pressure.

The endotoxin unit concentration (EU/m^3^) was assessed as described earlier [18]. The filters were placed in sterile 50 mL Greiner^®^ tubes (Greiner Bio One, Alphen aan den Rijn, Netherlands) with 4 mL pyrogen-free water containing 0.05% Tween-20. The tubes were placed in an end-over-end roller for one hour and centrifuged for 15 min at 1000× *g*. The supernatant was stored at −20 °C in 0.1 mL aliquots. The extracts were analyzed using a kinetic chromogenic Limulus amoebocyte lysate assay (Lonza, Breda, the Netherlands) in a dilution of 1:500. A 13-point standard curve ranging from 25 to 0.006 EU/mL was included in the assay as a reference.

#### 2.3.2. Ammonia

The ammonia concentration (ppm) was assessed with the use of Radiello™ *ready-to-use* diffuse samplers (Instituti Clinici Scientifici Maugeri, Pavia, Italy) pre-assembled with absorbent cartridges within the diffuse bodies, which binds ammonia in the form of ammonium, as described elsewhere [19]. The ammonia samplers were protected from urine and fecal contamination by an open bottom plastic casing. Until further processing, the ammonia cartridges were stored in closed zip-lock bags and cooled at 4 °C.

After extraction with 10 mL of deionized water, the samples were analyzed by a chemical colorimetric method based on the Berthelot reaction [20]. A standard curve ranging from 0.5 to 10 ng/mL ammonium was included in each assay as a reference to determine the amount in the air samples.

#### 2.3.3. Fungal Aerosols

Fungal aerosols were measured by active and passive sampling methods. First, for the active sampling method, D5600 Wuppertal pumps (Gebr. Becker^®^, Wuppertal, Germany) with a preset flow rate of 28.3 L/min were used. The pumps were connected to a Anderson one-stage 400-hole impactor (SKC Inc. Procare, Groningen, the Netherlands) equipped with Dichloran Glycerol 18% agar plate (DG18) and activated for a duration of 10 min. Second, for the passive sampling method, DG18 agar plates were placed directly in front of the enclosures for 10 min. In addition, to sample potential fungal spores originating from the ventilation system, DG18 agar plates were placed in front of the air inlets for 10 min. 

Subsequently, all samples were transported to the laboratory and were incubated at 24 °C. At 24, 48 and 96 h of incubation, the colonies were counted with a colony counter (Gallenkamp^®^, Loughborough, UK). The numbers of colonies on the agar plates were corrected by a Positive-Hole Correction table [21]. Moreover, the fungal colonies were microscopically identified to genus. The results were expressed as the number of colony forming units per cubic meter air (CFU/m³) and colony forming units per plate (CFU/plate) for the active and passive sampled plates, respectively.

#### 2.3.4. Temperature and Relative Humidity

Temperature (°C) and the relative humidity (%) were recorded using EL-MOTE-TH Temperature & Humidity Cloud-Connected Data Loggers (Lascar electronics^®^, Wiltshire, UK). The left compartments were provided with a datalogger, and one logger was placed outside on the BPRC premises to register the outdoor temperature and humidity, with a recording interval of ten minutes. The means of these recording intervals were calculated and used for further analyses. The dataloggers in the compartments were placed into the same protective boxes as the GilAir-5 pumps. For obvious reasons, the boxes were removed on cleaning days.

#### 2.3.5. Smoke Test

Non-toxic smoke was used to visualize the airflow in the animal enclosures. A pyrotechnic smoke cartridge Miniax KS (Scan-Air, Mill, the Netherlands) was lit in front of every air inlet in the indoor compartments. The distribution and flow of the smoke was recorded with cameras until the smoke was not visible anymore. All animals were locked in their outside enclosure during this test. Figure 3 shows the expected airflow.

#### 2.3.6. Personal Exposure

One animal caretaker per unit wore a GilAir-5 pump during a regular working day. The pump was attached to a waist belt, and the sampling head was attached to the collar of their coveralls to sample air as close to the mouth region as possible. In addition, caretakers kept a log of the tasks and the duration of these tasks that specific day. The measurements were paused when the caretakers left the unit for coffee and lunch breaks.

#### 2.3.7. Data Analyses

Statistical tests were performed with R studio v4.1.3 and GraphPad prism v8.4.2.

The dust, endotoxin, ammonia and fungal concentrations are presented as the GM with geometric standard deviation (GSD). GSD is defined as a multiplicative factor describing the range in a lognormal distribution used with GM, e.g., GM times or divided by GSD [22]. The between-unit and group differences for the dust, endotoxin, ammonia, temperature and relative humidity were tested non-parametrically using the Mann–Whitney U test. Non-parametric Spearman’s rank correlations were used to evaluate possible associations between dust, endotoxin, ammonia, temperature and relative humidity, between and within the units. 

Due to extreme precipitation during one night measurements, the correlations for temperature and relative humidity were also analyzed excluding night measurements. In addition, we investigated the occupancy rate and total body mass between and within the units and the IAQ parameters/contaminants. Furthermore, possible associations between sneezing, observed animal activity and the influence of days after high-pressure cleaning and the IAQ parameters were investigated. The number of sneezes and activity was evaluated with only the day measurements since the observations were performed during daytime. P values smaller than 0.05 were considered as statistically significant. Due to the limited data regarding fungal aerosols, statistical analysis was not performed.

## 3. Results

An overview of the analyzed associations between inhalable organic dust, endotoxins and ammonia is presented in Table 3. Additional associations between these parameters and other determinants are also shown in Table 3. Temperature and humidity correlations are presented in Table 4.

### 3.1. Dust, Endotoxins and Ammonia

The results of the stationary inhalable dust, endotoxin and ammonia measurements are presented in Figure 5.

The concentrations in left and right compartments of the units together were correlated (dust r_s_ = 0.59, *p* < 0.01; endotoxins r_s_ = 0.90, *p* < 0.001; and ammonia r_s_ = 0.61, *p* < 0.001). The GM dust concentration during daytime, in both RMU 0.069 (2.11) mg/m³ and CMU 0.068 (1.36), was significantly higher than during the nighttime, 0.033 (1.46) mg/m³ and 0.032 (1.31) mg/m³, *p* < 0.001, Mann–Whitney.

The GM endotoxin (EU/m³) concentration in RMU, 24.8 (1.81) EU/m³, was significantly higher in comparison to CMU, 6.44 (1.88) EU/m³, *p* < 0.001, Mann–Whitney. The results for the personal exposure of the caretakers are shown in Table 5. In addition, the time spent both inside and outside the animal rooms, as a percentage of the total sampling times, is presented. The caretakers were exposed to a 26–50 and 34–140 fold higher dust exposure compared to the GM concentration in the animal enclosures in RMU and CMU, respectively. Similarly, a 14–37 and 24–35 fold higher endotoxin exposure was observed.

Although a subjective observation, an ammonia odor was perceived in some enclosures, and on occasion, even a sharp ammonia odor was experienced by the observers. The measured ammonia levels in RMU, GM 1.84 (1.49) ppm, were overall higher compared to CMU, GM 0.58 (1.79) ppm, *p* < 0.001. Two outliers during the day and one in the night were reported in RMU as well as one outlier in CMU. On all these occasions, the simultaneously obtained results in the adjacent compartment were lower, suggesting fecal or urine soiling near the equipment.

A significant positive correlation was observed between the combined left and right endotoxin concentrations and ammonia levels (*r_s_* = 0.66) *p* < 0.001 Spearman’s rank. The dust concentration was correlated with the endotoxin concentration (r_s_ = 0.27, *p* < 0.05, Spearman’s rank. In addition, no association between days after high-pressure cleaning and dust concentration, endotoxin concentration and ammonia levels was observed. The interval between the cleaning procedure and the measurements ranged from 2 to 28 days. A correlation was observed between the body mass (kg) per m^3^ and both ammonia (r_s_ = 0.65, *p* <0.001, Spearman’s rank) and endotoxin (r_s_ = 0.76, *p* < 0.001, Spearman’s rank) concentrations.

### 3.2. Fungal Aerosols

The results of the active and passive sampling of fungal aerosols are summarized in Table 6. A total of 13 different fungi genera were observed; (1) *Paecilomyces* sp., (2) *Cladosporium* sp., (3) *Penicillium* sp., (4) *Aspergillus glaucus*, (5) *Wallemia* sp., (6) *Scopulariopis fusca*, (7) *Aspergillus ochraceus*, (8) *Aspergillus sydowii*, (9) *Aspergillus candidus*, (10) *Dydimella* sp., (11) *Alternaria* sp., (12) *Aspergillus niger* and one white sterile fungal colony that could not be further specified with only the light microscope. The first seven genera were the most dominant growing fungi. In addition, yeasts with a glistening pink or cream-colored appearance were observed; yet, it was not possible to specify these yeast colonies with the light microscope.

### 3.3. Temperature and Relative Humidity

The outdoor temperature ranged between 13.6 and 32.5 °C, and the relative humidity ranged between 60% and 100%. The indoor temperature in RMU ranged between 20.1 and 28.3 °C and the relative humidity between 51% and 91%. The indoor temperature in CMU ranged between 21.7 and 27.4 °C and the relative humidity between 49% and 79%. No significant differences were observed between the indoor temperatures of Group 1 and 2 within and between both units. 

However, a significant higher indoor relative humidity was observed in Group 1 compared to Group 2 in both units (*p* < 0.05). The indoor temperature was positively correlated with the indoor relative humidity (*r_s_* = 0.513, *p* < 0.050) when the night measurements were excluded. The absence of a correlation between these parameters when day and night are combined is due to a night measurement with heavy precipitation (Appendix B). The outdoor and indoor temperature were positively correlated as well (*r_s_* = 0.932, *p* < 0.001).

The night measurements were not included to calculate correlations between the indoor temperature and relative humidity and the air quality parameters. In both units, a negative correlation was observed between the indoor temperature and the endotoxin concentration r_s_ = −0.54, p < 0.01 (Spearman’s rank) and r_s_ = −0.59, *p* < 0.01(Spearman’s rank) for RMU and CMU, respectively. The relative humidity and the dust concentration were positively correlated in RMU (r_s_ = 0.48, *p* < 0.05, Spearman’s rank) but not in CMU. For CMU alone, a negative correlation between relative humidity and endotoxins was observed. In addition, no correlation was observed between both the indoor and outdoor temperature and relative humidity and ammonia concentrations.

### 3.4. Smoke Test

#### 3.4.1. RMU

Figure 6 provides a schematic cross sectional view of the units presenting the present airflow visualized with the smoke test. Immediately after lighting a cartridge, the smoke went through the fencing of the enclosure, along the ceiling into the compartment. The smoke descended after it collided against the back wall, causing the smoke to reach the floor. Next, the flow rate slowed down, and the smoke diffused in the compartment and remained for approximately six minutes before moving slowly towards the ventilation-outlets. 

The largest part of the smoke departed through the upper outlet. The total duration from the production until the disappearance of the smoke took approximately eight minutes. Overall, similar patterns were seen in the different compartments with the exception of the right compartment of Group 2, where the smoke moved to the ventilation outlets along the ceiling, without dispersal in the room.

#### 3.4.2. CMU

In contrast to the RMU, the smoke was largely stopped by the fence framework of the left and right compartment of both groups (Figure 5). While most of the smoke stayed in the corridor, some smoke entered the compartment by passing along the ceiling. The remainder of smoke that reached the back wall of the enclosure descended slowly downwards. Eventually, it distributed through the whole space, except for the floor. A stationary layer of smoke was formed about 50–100 cm above the bedding (Figure 5). There was no movement of smoke for approximately 15–20 min. Finally, the smoke left the enclosure across the ceiling through the upper outlet of the ventilation system, and after 20–30 min the smoke was not visible anymore.

In Group 2, the strip curtain in the left compartment was damaged and acted as an open connection to the outdoor enclosures. The smoke that moved along the ceiling disappeared partially yet quickly through the curtains to the outdoors. The remainder of the smoke reached the back wall, descended along the wall and formed a stationary layer 80 cm above the floor. Due to the loss of smoke, this layer was less apparent and was not visible anymore after approximately 15 min.

### 3.5. Number of Sneezes

The total number of sneezes over the five study days was 3.5 and 6.5 for RMU Groups 1 and 2, respectively, and 1.3 and 5.6 for CMU Groups 1 and 2. This was calculated as the sum of the observed number of sneezes during the five minute time-frames and corrected for the number of animals present in the two compartments during this time-frame. No significant differences were observed between the compartments or units. No correlations were observed between the total number of sneezes and air temperature or relative humidity. The activities displayed by the macaques during the observation of sneezing were mainly foraging and playing in the morning and resting and a little foraging in the afternoon. No correlations were observed between the total numbers of sneezes and dust, endotoxin and ammonia concentrations. Furthermore, no correlation was observed between the total numbers of sneezes and the time after cleaning of the enclosures.

## 4. Discussion

This study aimed to assess the IAQ of group-housed rhesus and cynomolgus macaques in a breeding facility. The acquired data of the IAQ parameters in the NHP enclosures, fungal aerosol concentrations combined with the smoke tests provided an informed state of the average air quality in both RMU and CMU. Additionally, the caretakers were exposed to higher inhalable organic dust and endotoxin concentrations compared to the animals in their indoor enclosures.

### 4.1. Inhalable Dust

There is no Dutch exposure limit for organic dust; however, the Danish occupational health council recommends an average daily limit of 3 mg/m^3^ exposure for organic inhalable dust [23]. Our results in the animal enclosures comply with the TLV, yet organic dust exposure to the caretakers exceeded the TLV during two days.

The dust concentration measured during the day was significantly higher compared to the night measurements. The most reasonable explanation is the absence of human and animal activity during the night, since the animals were sleeping on platforms above the floor in the indoor enclosures. Similar to our observations, a strong correlation between animal activity and dust concentration (PM_10_) was observed in pigs [24]. Furthermore, the results in RMU and CMU are comparable to previously reported dust concentrations in conventional laboratory rabbit rooms, i.e., 0.06 and 0.07 mg/m^3^ in macaques compared to 0.1 mg/m^3^ in rabbit rooms [25]. 

In addition, a negative correlation for relative humidity (range 42–68%) and dust had previously been observed [24]. During scheduled spraying sessions in a pig barn, a reduction in dust levels was observed [24]. Yet, our results revealed a positive correlation between these parameters in RMU. Our lower mean indoor temperature and dust concentration as well as differences in ventilation, species and housing could have contributed to these contrasting findings.

Work activities in RMU and CMU resulted in the higher exposure of caretakers to dust compared with work activities in rabbit rooms: 1.83–9.28 mg/m^3^ compared to <0.5–2.3 mg/m^3^, respectively [25]. It was previously reported that, apart from differences in activities, variation in the dust content of clean bedding materials may contribute to this difference as well. Reported dust concentrations (mg/m^3^) of Lignocel^®^ 3–4 were 3.5–13 times higher compared to Tapvei 4HP bedding [26].

Although the dust exposure of the caretakers was variable, the wearing of personal protective equipment (e.g., FFP 2 masks or N95) is advisable and was already mandatory at our facility. Therefore, the actual inhaled concentrations of organic dust should be lower than the exposed concentrations.

### 4.2. Endotoxins

For endotoxins, the health council of the Netherlands has recommended two health-based exposure limits: 90 EU/m^3^ for the occupational population and 30 EU/m^3^ for the general population [27]. Both were exceeded at the BPRC. Whereas the value limit of 30 EU/m^3^ was exceeded at several but not all occasions of stationary measurements in RMU and CMU, the results of the personal measurements exceeded the occupational limit of 90 EU/m^3^ on all occasions.

In the current study a significantly higher concentration of endotoxins was observed in the RMU compared to the CMU. Although the roof air handling units were not identical regarding the manufacturer and age, it is unlikely that this caused the observed difference in endotoxin concentrations between RMU and CMU since the dust concentrations were similar. Possibly, the more efficient air circulation in RMU, as revealed by the smoke test, added to this result in endotoxin difference. However, the ammonia concentrations were not influenced by this, most likely because gaseous substances diffused more easily in the entire animal room and to the outdoors.

Furthermore, the occupancy rates were similar between the units, though a correlation was observed between body mass per m^3^ and endotoxin concentrations. Due to the limited number of observations, statistical analysis was not performed separately for the rhesus and cynomolgus macaques. However, the cynomolgus macaques had a lower mean bodyweight compared to the rhesus macaques (Table 1). This finding is in line with an earlier report describing a higher weight-to-height index for rhesus macaques compared to cynomolgus macaques [28]. Therefore, we assume that heavier animals produce more excreta, which, in turn, facilitates more bacteria and endotoxin.

The observed correlation between indoor temperature and endotoxin concentration could be explained by an expected lower indoor occupancy rate during hot days. Although the number of animals were recorded for a total of two hours per study day, these data were not sufficient to link the indoor occupancy to the outside weather conditions. The absence of animals and a decrease of fecal and urine soiling in the indoor enclosure could have resulted in a decrease in the endotoxin concentration during sunny days. However, the reason for the observed negative correlation between relative humidity and endotoxins in CMU, other than a limited number of observations, remains unclear.

The endotoxin concentration in rabbit rooms ranged from 10 to 13 EU/m^3^ (1 ng = 10 EU), while we measured 7.48–19.08 EU/m^3^ [25]. Personal exposure ranged between 7 and 36 EU/m^3^ in the rabbit rooms and between 165 and 969 EU/m^3^ in our CMU and RMU. Furthermore, workers in horse stables with wood chip bedding were reported to have been exposed to GM of 742 EU/m^3^ during eight hour work shifts [29]. These data suggest a higher exposure when working in macaque species enclosures. However, the maximum measured exposure was 969 EU/m^3^ for macaque caretakers compared to 9846 EU/m^3^ for horse caretakers. Compared to the presented data in rabbit rooms and horse stables, working in macaque units resulted in the same order of magnitude of personal exposure to endotoxin concentrations.

Sweeping the floor in horse stables was identified as being responsible for the predominant endotoxin exposure [17]. The authors proposed, as a preventive measure, to wet the surface prior to sweeping. In NHP facilities, this preventive measure is not considered reasonable as it would make the cleaning physically too heavy to perform for the caretakers. Others have proposed the use of vacuum systems, yet, such a system must be powerful enough to vacuum large quantities of soiled sawdust in order to maximize the exposure reduction for our caretakers [30].

However, our data showed that both the health-based and agricultural endotoxin TLV were exceeded during daily work activities. FFP 2 masks are known to protect against dust, viruses and bacteria, and a 10-fold reduction to endotoxin exposure while wearing these masks was estimated [31]. The efficiency of a facemask is not only influenced by characteristics of the dust particles, i.e., the aerodynamic size, but also by the fitting characteristics [32,33]. Although we do not know the size distribution of the endotoxins observed in the macaque units, it is reasonable that wearing good, fitted facemasks should reduce the exposure.

### 4.3. Ammonia

Our results did not reveal a correlation between both the indoor and outdoor temperature and humidity and ammonia concentrations. Furthermore, the filters used in the air handling unit are not effective against volatile organic compounds, such as ammonia.

Our data revealed a GM ammonia concentration of 1.02 ppm. The reported odor threshold of ammonia in humans is approximately 0.05–5 ppm, and the irritation threshold 31–314 ppm [34,35,36,37]. Ammonia was smelled by the observers in some of the enclosures. However, the variety in the reported odor threshold and irritation threshold suggest that personal observations are not reliable to detect high ammonia levels.

To the authors’ knowledge, there are no data available regarding ammonia concentrations measured with a direct method in NHP facilities. Therefore, it is not possible to compare our results with others.

In the current study, the ammonia samplers were protected from soiling by an open bottom plastic casing. Nevertheless, the animals were able to be in close proximity with the measuring equipment. Most likely, urine or fecal contamination near the samplers caused the reported outliers. This assumption is supported by the observed, lower, ammonia concentrations in the corresponding compartments. For example, in the left compartment during the night measurement, a value of 6.2 ppm was observed compared to 0.61 ppm in the right compartment.

The ammonia levels in laboratory rabbit rooms (0.14–0.29 ppm) were lower compared to our ammonia levels, which is likely due a difference in the bedding change frequency: twice a week in rabbit rooms compared to once a week in NHP [25]. In addition, the concrete floors of the RMU and CMU are more porous than the trays in the rabbit rooms and therefore possibly provide a temporarily reservoir for micro-organisms and animal waste. A higher cleaning frequency or the application of a non-porous top layer in the RMU and CMU could result in a further decrease of ammonia concentrations.

Ammonia levels for livestock are higher than our observed ammonia levels. Mean ammonia levels of 5–18 ppm in pig barns, 5–30 ppm in poultry barns and <8 ppm in cattle barns were described [38]. Moreover, for pigs, it was described that exposure to 80 ppm of ammonia induces an increase in thickness of the nasal mucosa [15]. In addition, pigs exposed to 50 ppm showed significantly increased serum urea and triglyceride concentrations [39]. Unfortunately, investigating the possible effects of the exposure of certain levels of ammonia on the mucosal thickness of the nose and blood serum concentrations were not included in our study. For NHP, no data are available regarding the subclinical effects of long-term exposure to relatively low levels of ammonia.

### 4.4. Fungi

Airborne fungal levels, measured by the active sampling method, were overall higher compared to previously reported levels in NHP facilities (median 71 CFU/m^3^, range 0–635) [40]. However, compared to monkey enclosures in a zoo setting, our results are well below their observed median (2929 CFU/m^3^) and range (2461–3294 CFU/m^3^) [41]. Occupancy rates (animals per m^2^) varied in these studies and included 1.2–3.6, 22.8 and 2.4–3.6 for the laboratory-housed monkeys, zoo-housed monkey and the monkeys housed at BPRC, respectively. These data support the hypothesis that airborne fungi are not primarily influenced by occupancy rates [40].

The primary source of fungi in animal enclosures is likely the bedding material [26,29,40]. Although neither of these studies mentioned the amount of used bedding material, it is plausible that the use of 31 kg of clean wood fiber bedding per compartment after each cleaning procedure contributed largely to the observed airborne fungi levels in this study. Furthermore, the ventilation rate may affect the fungal aerosol levels. Only in the zoo, the setting of air exchange for six times a day was reported [41]. 

Compared to the ventilation rate in both RMU and CMU of six exchanges an hour, this substantially higher frequency could be an explanation for the observed differences in fungal aerosols levels. Overall, most of the observed fungal species were comparable to the earlier reported genera in monkey and rabbit laboratory rooms [25,26,40]. Although these species are seldom the cause of primary infection, the Finnish government states that people should not be exposed to indoor levels exceeding 500 CFU/m^3^ [42,43,44]. However, other guidelines vary greatly and range from less than 100 to over 1000 CFU/m^3^ of total aerosol fungal concentrations [45]. 

Non-contaminated indoor fungal concentrations are mostly less than 1000 CFU/m^3^ [46]. High concentrations of fungal aerosols, e.g., >1000 CFU/m3 were associated with animal handling [45]. Although personal exposure was not investigated in our study, it is assumable that the TLV are exceeded regularly—in particular on cleaning days. However, a N95 mask filters 95% of the particles to at least 0.3 µm [32]. A size distribution of fungal aerosols ranging from 0.65 to 11.0 µm was reported [41], suggesting that a regular FFP2 mask should provide protection against fungal inhalation.

### 4.5. Temperature and Relative Humidity

The thermoneutral zone for rhesus macaques is 24.7–30.6 °C, while a range between 16 and 25 °C is considered to be appropriate for macaque species and, in particular, 21–28 °C regarding cynomolgus macaques [47,48]. Our indoor temperature measurements ranging from 21.7 to 27.4 °C should thus be considered as comfortable for the macaques at BPRC.

In general, a relative humidity of <30% is considered low, and a relative humidity of >80% as high. These are not absolute values, depending on other factors—for example, climate [49]. In addition, it was described for human subjects that low relative humidity (<30%) results in dryness of the ocular mucosa and skin and eventually dryness of the nasal mucous membranes [50]. Similarly, an increased wetting length of the Schirmer tear test with increased relative humidity for dogs was demonstrated [51]. In our study, we observed a high relative humidity (>80%) only once in RMU during a night measurement while there was heavy precipitation outside (Appendix A). Despite the fact the BPRC has no humidity control system, the relative humidity values observed were mostly within the comfortable ranges, e.g., a range of 51–91% in RMU and 49–79% in CMU.

### 4.6. Smoke Test

The U-channels used at the BPRC to protect the operating mechanism of the hatches inside the enclosures CMU had an unexpected negative impact on the airflow (Figure 5). Although the IAQ parameters were within human TLV in the animal enclosures, adjustments to the fencing or adjustments to the air inlet in CMU are advisable in order to optimize the ventilation in the enclosures. One should keep in mind that, next to fences, all other provided cage constructions can have a negative influence or even block the airflow [52,53]. Although methods, such as computational fluid dynamics, are more sophisticated to visualize airflow, smoke cartridges are a relatively easy and cheap method to reveal major flaws in air circulation.

### 4.7. Sneezing

Poor IAQ can cause several respiratory symptoms, both acute and chronic, including chemical- or inflammatory-induced itchy eyes, runny nose, sneezing and coughing [54,55,56]. We observed no correlation between the total number of sneezes and the IAQ parameters or duration after cleaning. The duration and frequency of our recordings, however, could have been too low to reveal this. Sneezing could also be an unsuitable parameter to predict the IAQ. However, the inhalable dust, ammonia and endotoxin concentrations were within acceptable ranges; thus, it was not expected to induce sneezing.

As an alternative to the sneezing recordings, an aversion observation could be performed. A strong animal preference for fresh air in an atmospheric ammonia preference test for pigs was described [57]. These animals spent significantly less time in ammonia atmospheres ≥10 ppm. It is possible that macaques show a similar aversion towards high ammonia concentrations in indoor enclosures. We hypothesize that we may observe an increased number of animals in the outdoor enclosures when a cleaning day approaches. As reported, forced choices, such as differences in the indoor and outdoor temperatures, should be avoided during these observations as pigs were shown to give priority to environmental temperature over ammonia concentrations [57].

### 4.8. Limitations and Recommendations

Although the data of this applied assessment of IAQ in group-housed macaques demonstrate the overall air quality, there are also some critical attention points. The main pitfall was the difference in the construction and design of the RMU and CMU. For example, in both groups in RMU the right compartment had no outdoor access. Although a moderate-to-strong correlation was observed between the left and right compartments of the combined species, the smoke test showed a deviating air circulation in the right compartment in RMU Group 2. Since the CMU had outdoor access in both compartments, the smoke test observations between the units were not directly comparable. Therefore, it remains unclear if the primary cause of the deviating smoke pattern in the right compartment of RMU Group 2 was indeed due to the absence of outdoor access. 

Personal exposure of the caretakers during their daily work routine is considered to exceed the TLV for inhalable dust, endotoxins and fungi. In addition, several studies demonstrated that cleaning activities induce increased exposure [4,8,40]. Therefore, it is advised to provide sufficient personal protective equipment and to increase the frequency of air changes during cleaning procedures. Despite the fact that most IAQ parameters were within human TLV in the animal enclosures, more research is recommended to investigate the subclinical effect of relatively low exposure to dust, endotoxins and ammonia on the respiratory tract of macaques. 

It would be interesting to research these chronic effects on nasal cytology, bronchoalveolar lavage cytology and clinical chemistry and hematology. However, in order to put the data in perspective, a control group is essential. As mentioned previously, an observational aversion test could be beneficial to establish animal preferences. Finally, prolonged stationary air sampling could possibly reveal more significant differences over time.

## 5. Conclusions

This study showed that the inhalable dust, endotoxin and ammonia concentrations in the NHP breeding facility did not exceed human TLV; however, the personal organic inhalable dust and endotoxin exposure of the caretakers exceeded the TLV. Moreover, our results may increase awareness of the IAQ, which may reduce caretaker exposure during daily work routines. In conclusion, we recommend to assess the IAQ in old and newly built NHP facilities to ensure the optimal IAQ for both animals and caretakers and to provide adequate personal protection materials for the caretakers.

## Figures and Tables

**Figure 1 animals-12-01750-f001:**
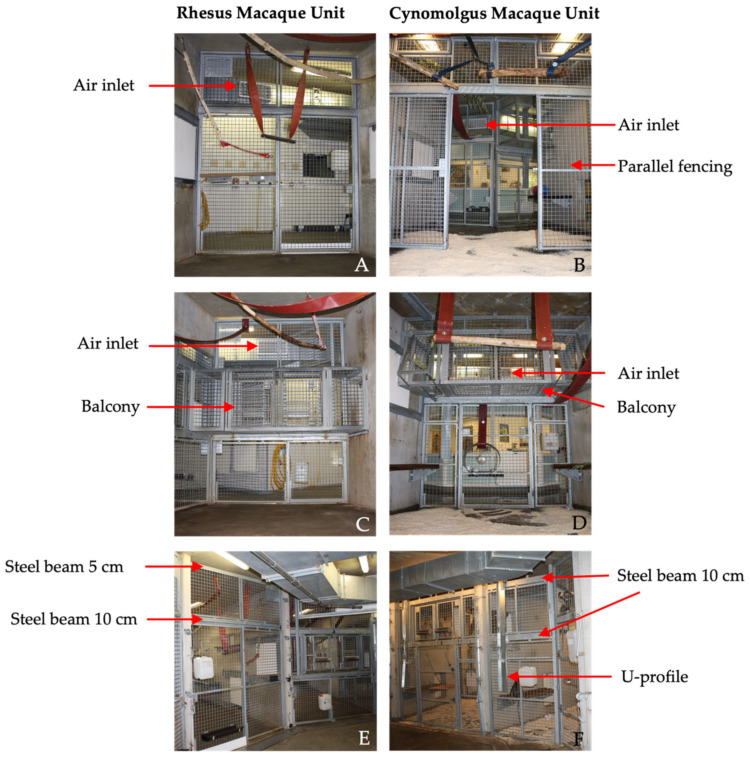
Overview of the enclosures of the Rhesus macaque unit (RMU) and Cynomolgus macaque unit (CMU). (**A**–**D**) The front fencing viewed from the inside of the enclosures. The air inlets in these views are visible in the background behind the fencing. (**B**) The parallel fence in CMU with open sliding doors is clearly visible. (**C**,**D**) The location of the balcony in both units. (**E**,**F**) A view from the caretaker hallway. The upper steel beam in RMU (**E**) is 5 cm compared to 10 cm in CMU (**F**). This beam is located 10 cm from the ceiling in RMU and 16 cm in CMU. (**F**) The additional U-profile on the front fence in CMU, designed to protect the control-wires from the cynomolgus macaques is shown.

**Figure 2 animals-12-01750-f002:**
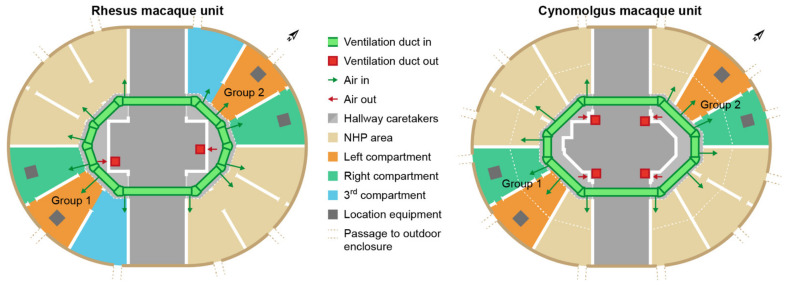
Schematic overview of the studied rhesus and cynomolgus macaque units. Both units consist of two separate animal rooms. In each room, the study group is highlighted. In the rhesus macaques unit (RMU), two of the three compartments are directly connected to the outdoor enclosures, whereas all compartments in the cynomolgus macaques unit (CMU) are connected to the outdoor enclosures. RMU consists of three and CMU consists of two indoor compartments per group. The compartments are separated with concrete walls with passageways for the animals.

**Figure 3 animals-12-01750-f003:**
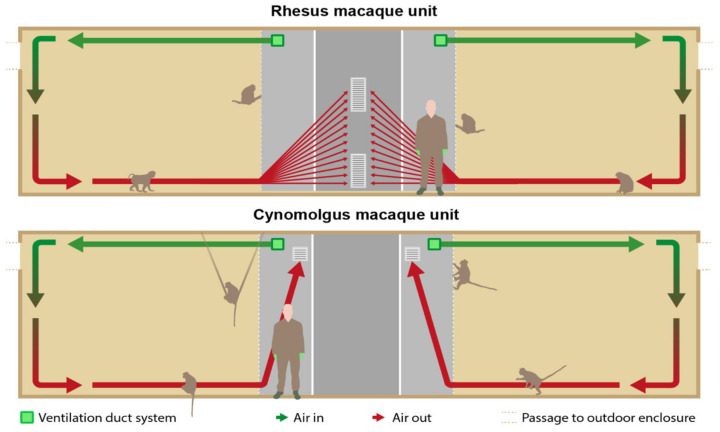
Schematic cross sectional view of the units presenting the intended airflow. In the rhesus macaque unit (RMU), the exhausts were located in the middle of the wall opposite to the cages and right under the ceiling and above the floor. In the cynomolgus macaque unit (CMU), the exhausts were located on the left and right side of each room right under the ceiling and above the floor and opposite to the cages as well.

**Figure 4 animals-12-01750-f004:**
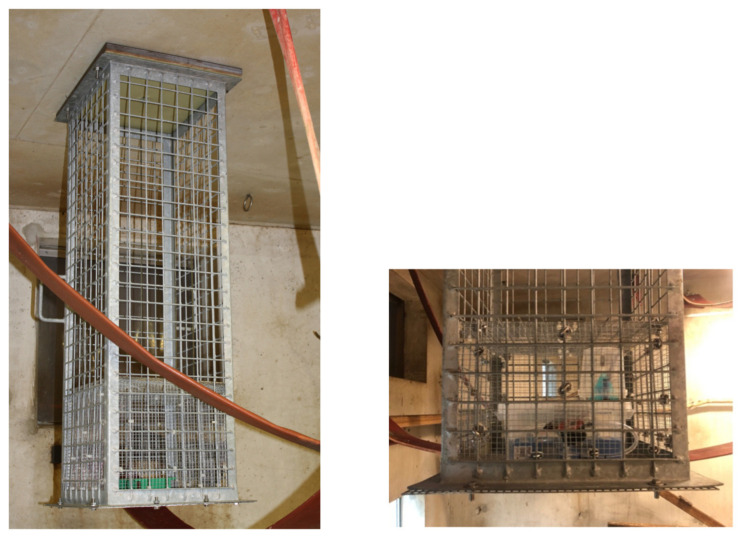
Experimental setup, the cage construction was tightly secured to the ceiling (left) and a close-up (right) of the additional compartment with smaller mesh wire to protect the measuring equipment.

**Figure 5 animals-12-01750-f005:**
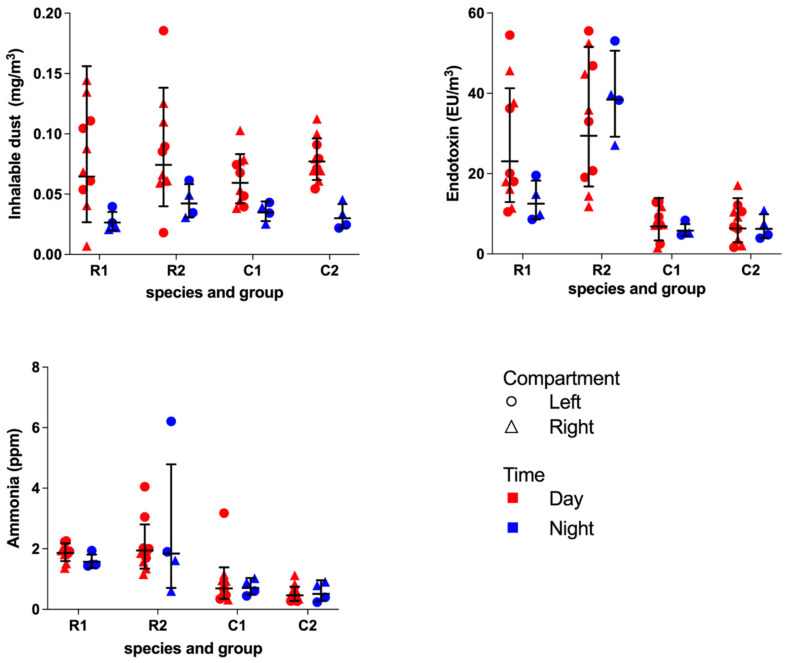
The results of the day and night measurements of inhalable dust, endotoxins and ammonia of each compartment of both groups in the rhesus macaque unit (RMU) and cynomolgus macaque unit (CMU) presented as individual measurements, geometric mean (GM) and geometric standard deviation (GSD).

**Figure 6 animals-12-01750-f006:**
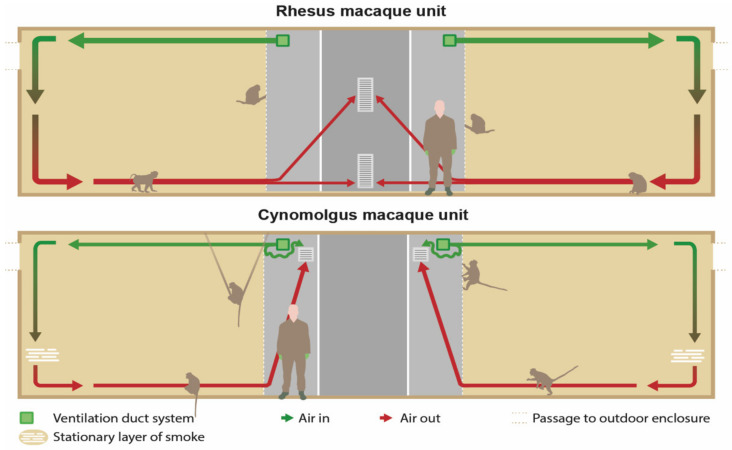
Schematic cross sectional view of the units presenting the present airflow visualized with the smoke test.

**Table 1 animals-12-01750-t001:** Descriptive characteristics of the animals (mean (min-max)) and enclosures of the four study groups.

	Rhesus Macaque Unit		Cynomolgus Macaque Unit	
	Group 1	Group 2	Group 1	Group 2
Age (years)	6.6 (0.3–19.1)	7.0 (2.1–18.2)	3.5 (1.8–12.1)	7.5 (0.5–26.6)
Weight (kg)	5.7 (1.3–13.6)	6.9 (3.5–12.2)	3.2 (2.3–4)	3.7 (1.1–9.2)
Number of animals (N)	23	33	14	21
Occupancy (N/m^3^)	0.11	0.1	0.1	0.15

**Table 2 animals-12-01750-t002:** Definitions of the selected animal activities.

	Activity	Definition
1	Foraging	The animal is positioned on the floor and is manipulating the wood fibers while looking for edible parts.
2a	Play terrestrial	Social play behavior, e.g., chasing, wrestling or solitary play, e.g., object play, on the floor of the enclosure.
2b	Play arboreal	Social play behavior, e.g., chase, climbing or solitary play with, e.g., object play displayed on platforms, beams and other enrichment items.
3	Rest	The animals are resting, sleeping or grooming, absence of locomotion anywhere in the enclosure.
4	Aggressive interaction	Aggressive behavior, e.g., attack, escape or give ground.

**Table 3 animals-12-01750-t003:** Correlation (Spearman’s rank) matrix between indoor air quality parameters (IAQ) in the left and right compartments and between the IAQ parameters during day and night measurements, differentiated for combined day and night, day only and night only of the combined species. In addition, correlation (Spearman’s rank) between IAQ and the different determinants are presented.

Air quality Parameters		Dust		Endotoxin		Ammonia	
		r-Value	*p*	r-Value	*p*	r-Value	*p*
Left and right compartments		**0.59**	**0.001**	**0.90**	**7.42** × 10^−^^7^	**0.61**	**0.0007**
Dust	Day and night			**0.27**	**0.04**	0.11	0.4012
Endotoxin	Day and night					**0.66**	**6.78** × 10^−8^
Dust ^a^	Day			**0.31**	**0.049**	0.13	0.10
Endotoxin ^a^	Day					**0.66**	**7.46** × 10^−6^
Dust	Night			0.39	0.14	0.11	0.67
Endotoxin	Night					**0.71**	**0.003**
**Bodyweight/m^3^**		0.13	0.35	**0.76**	**1.01** × 10^−11^	**0.65**	**6.22** × 10^−8^
**Groups within**	RMU	**0.56**	**0.038**	0.44	0.12	0.41	0.15
	CMU	**0.68**	**0.0095**	**0.65**	**0.014**	0.39	0.7
**Indoor Temp ^a^**	RMU	0.22	0.35	**−0.54**	**0.014**	−0.36	0.11
	CMU	0.03	0.90	**−0.59**	**0.0066**	−0.27	0.26
**Indoor RH ^a^**	RMU	**0.48**	**0.032**	−0.30	0.19	−0.12	0.61
	CMU	0.01	0.97	**0.73**	**0.00025**	0.08	0.74
**Number of sneezes**		−0.18	0.46	0.02	0.92	0.10	0.66
**Days after cleaning**		−0.03	0.83	−0.19	0.15	−0.13	0.34

Significant correlations are shown in boldface. ^a^ Nights excluded.

**Table 4 animals-12-01750-t004:** Correlation (Spearman’s rank) matrix for the temperature (Temp) and relative humidity (RH) and determinants.

	Indoor Temp		Outdoor Temp		Indoor RH		Outdoor RH	
	r-Value	*p*	r-Value	*p*	r-Value	*p*	r-Value	*p*
Outdoor Temp	**0.82**	**7.63** × 10^−^^15^						
Indoor RH	0.11	0.41	0.10	0.49				
Outdoor RH	**−0.27**	**0.04**	**−0.53**	**2.10** × 10^−^^5^	**0.47**	**0.0003**		
Number of sneezes ^a^	−0.10	0.68	−0.10	0.71	−0.08	0.75	0.0003	0.82

Significant correlations are shown in boldface. ^a^ Nights excluded.

**Table 5 animals-12-01750-t005:** Measured personal exposure of caretakers to inhalable dust and endotoxins during two routine working days. The duration of the tasks performed by the caretakers both inside and outside the animal rooms are presented as a percentage (%) of the total sampled time per working day. In addition, the GM concentrations of inhalable dust and endotoxins during the stationary day measurements in each unit are presented.

	Rhesus Macaque Unit		Cynomolgus Macaque Unit	
	Day 1	Day 2	Day 1	Day 2
Inhalable dust (mg/m^3^)	3.3	1.8	2.3	9.3
Endotoxins (EU/m^3^)	390.7	968.6	164.7	231.7
Total sample time (min)	357	362	389	393
Time inside animal room (%)	66%	52%	45%	39%
Time outside animal room (%)	34%	48%	55%	61%
GM stationary inhalable dust in unit (mg/m^3^)	0.069		0.067	
GM stationary endotoxins (EU/m^3^) in unit	26.1		6.61	

**Table 6 animals-12-01750-t006:** Airborne fungi measured by the passive and active sampling method (median (min–max)), presented in colony forming units (CFU/m3 or CFU/plate, respectively).

	Rhesus Macaque Unit		Cynomolgus Macaque Unit	
Group	1	2	1	2
Active sampling method (*n* = 3)				
Left compartment (CFU/m^3^)	464 (425–1877)	836 (653–1113)	566 (430–659)	593 (479–884)
Passive sampling method (*n* = 2)				
Inlet (CFU/plate)	11	6	4	11
	3	3	2	7
Floor (CFU/plate)	5	36	9	12
	5	34	22	26

## Data Availability

Data are available on reasonable request.

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
