# Peer review of "Assessment of Indoor Air Quality for Group-Housed Macaques (Macaca spp.)"

_animals, 2022, doi:10.3390/ani12141750_

Round 1

Reviewer 1 Report

This reviewer would like to thank the authors for their submission.  This will provide data in an important area that has few objective reports in the literature.

Introduction

Line 64 - “European directive” is not cited.  It is assumed that this is European Directive 2010/63/EU.

Line 64 - the citation for the Guide for the Care and Use of Laboratory Animals refers to a book review, not the original source.  Is that intentional?

Materials and Methods

Line 102 - AAALAC International is the newly accepted title of this organization.  The organization no longer uses the entire name of Association for the Assessment and Accreditation of Laboratory Animal Care.

Line 114 - “Surface of 25m2.”  Please specify clarify if this is floor area or something different.

Figure 1 caption - the RMU design is different from that of the CMU, in that it is 50% larger and one of the sampling rooms does not have outdoor access.  This may have significant effect on the results, but is not covered in much detail in the discussion.

Line 161 - ventilation rate is considered sufficient due to access to the outdoors, but there is no demonstrated basis for 6 air changes per hour.  How was this exchange rate determined, and given the results of the manuscript, is it still considered sufficient?  What percentage of time did animals spend outdoors during the sampling period, and did this differ between rhesus and cynomolgus macaques?

Line 172 - Was there a difference in the interval since the last cleaning of these enclosures for the different rooms or the RMU vs. the CMU?  If the RMU and CMU were sampled simultaneously, did this allow a broad range of times since cleaning in different rooms?  It is later shown that this interval did not correlate with sneezing, dust, endotoxin, or ammonia, but it is not clear that this interval was sufficiently assessed.  Do the sampling days cover the entire range between cleanings?

Line 173 - dust, endotoxin, and ammonia samples were collected during five separate days.  These are sequential days, correct?  Or were sampling days decided randomly or designed to provide representation of differing intervals since the last washing of cages?

Line 190 - why were dam-infant pairs counted as one while infants were carried?  If the infant is not being carried, do they then count as two?  Does this count match what is depicted in Table 1?

Line 192 - should “non-evasive” be “non-invasive”?

Line 205 - while samples were weighed in an acclimated room, were all samples (pre- and post- and each sampling location) weighed at the same time?  Could there be variation based especially on humidity, but also temperature and atmospheric pressure?

Line 207 - “Spaan et al. (2008)” reference is not provided.

Line 207 - is there a positive control for the endotoxin assay?  Is a curve obtained using a known standard?

Line 210 - does the phrase, “over one hour,” mean greater than one hour or over the course of one hour (i.e. 60 minutes)?  If it is greater than one hour, was there variation between samples?

Line 226 - fungal sampling apparatus.  Is one D5600 pump connected to on impactor, or are both pumps attached to the impactor?  Was this repeated at different times, especially since the last washing of the rooms?  Table 4 shows n=3 for active sampling and n=2 for passive sampling.  These were also simultaneously, as described above, correct?

Line 236 - number of CFU/m3.  Was there a cutoff for fungal particles based on the 400-hole impactor to avoid overloaded samples?

Line 242 - temperature and relative humidity were sampled continuously.  How were these data analyzed compared to the IAQ parameters, which are more of a discrete value?  Perhaps this is best addressed in Section 2.3.7. Data analyses.  It is not clear if a discrete temperature is used in analysis to coincide with the IAQ data, but continuous data are represented when talking only of temperature and humidity.

Line 352 - similar to above comments, Lines 172 and 173.  The time interval since the last cleaning is not presented, only an r-value and p-value in Table 2.b.

Line 404 - the right compartment of group 2.  This is the compartment that has no outdoor access, correct (e.g. right side as the observer is facing the enclosures)?  Does this difference lead to less mixing of room air?  Are RMU groups 1 and 2 different, in that group 1 has more mixing due to the orientation of the outdoor access door to prevailing winds or some other factor?

Line 540 - is this a sentence fragment?  The entire paragraph consists of one sentence that ends in a dash.

Line 643 - “where” should be “were.”

Reviewer 2 Report

This paper presents the first evaluation of air quality in a non-human primate housing facility and relates the findings to published threshold values for a range of air quality measures. The study is important in providing the first such evaluation and opens the door for further research into this as-of-yet unexplored aspect of animal colony management. The manuscript is well-written for the most part and there are a few minor language errors. The authors would do well to ensure that the motivation for all of the methods used is clear and that the associated descriptions are also made clear to the reader (see comments on behavioural observations). The authors should also ensure that the methodological descriptions are done in a manner which is logical and allows the reader to understand without having to retrospectively construct information into a whole picture.

With these relatively minor issues addressed, I see no reason why this paper should not be considered for publication. As such I am recommending that the publication of this study go ahead following minor revision.

Reviewer 3 Report

Overall impression

This manuscript focuses on an important but underrepresented topic of animal health in laboratory settings. With edits, this review could contribute to advances in our knowledge of how indoor air quality can affect caretakers and animals. 

The ideas and intention behind the project are good, as the responses of animals in human care to disruptive events definitely require further study. As zoo researchers, the authors have done a good job designing a data collection scheme for a small sample size. My overall impression is that this cursory review of measures for air quality was sufficient in identifying gaps and considerations which must be further investigated. There was significant amount of literature presented, and the content matches the goal of the study. Likewise, the authors break the content into easily chewable pieces which flow well. 

Unfortunately, as written, the manuscript requires editing. I suggest another pass to fix incomplete sentences, instances of improper grammar, and several errors in punctuation. I have highlighted some of these below, but did not include an exhaustive edit. 

Introduction 

The authors provide a succinct justification and outline of the study, with relevant background information. 

Lns 83-85: is a good example of a sentence which requires more context to be clearly understood, e.g., "...must be improved in research environments" or "...must be improved for animal models" or the like. 

Lns90-91: recommend listing both species names here

Methods 

Please edit this section for clarity on how the study was conducted. As written, several tests would not be reproducible. 

Lns 113-121: since RMU and CMU had differing effects due to materials or construction of enclosures, I recommend more detail here - solid wall or mesh, type of mesh (chain link/cyclone mesh, welded wire) and size. Often CMU are held behind mesh with a tighter weave, but in this study it would be important to know if that is true here or not. 

Regarding behavioral assessment, more detail is needed. For instance, authors note behavioral data was collected in four sessions 30min on study days, and air quality measurements were taken 6 hours a day on each study day. Were the 30min sessions distributed evenly or randomly throughout the 6 hours study period, and were observations taken live or recorded, what data collection method(s) used (instantaneous, ad lib, etc). On page 13, authors note using "5 minute time frames", however such time frames are not mentioned here.  

Lns187-188: unsure what "...subsequent group observations were alternately performed" means. 

2.3.7 Data analysis: authors note the majority of the data was skewed. If data was logtransformed to obtain normality, please justify why nonparametric statistics were used. 

Discussion

Like the Introduction, Methods, and Results, the layout of the Discussion aides the reader in following the multiple tests conducted, and subsequent arguments made. 

Lns4664-467, Another example of editing: The first sentence mentions rabbit rooms, but the reference to rabbit rooms is not made until the second sentence, so as written, its confusing.  Recommend re-writing this paragraph. 

Lns539-540: The punctuation here doesn't make sense, so assume the hyphen is a keystroke error.  

Round 2

Reviewer 3 Report

Thank you for the revisions to the manuscript and attention to my thoughts. I have no additional comments, and recommend to accept this manuscript for publication. 

Author Response

Dear reviewer,

Thank you very much for your recommendation to publish our manuscript!

Kind regards,

Annemiek Maaskant